# Prevalence of Variants of Uncertain Significance in Patients Undergoing Genetic Testing for Hereditary Breast and Ovarian Cancer and Lynch Syndrome

**DOI:** 10.3390/cancers15245762

**Published:** 2023-12-08

**Authors:** Pavlina Chrysafi, Chinmay T. Jani, Margaret Lotz, Omar Al Omari, Harpreet Singh, Katherine Stafford, Lipisha Agarwal, Arashdeep Rupal, Abdul Qadir Dar, Abby Dangelo, Prudence Lam

**Affiliations:** 1Department of Medicine, Mount Auburn Hospital, Cambridge, MA 02138, USA; pavlina.chrysafi@mah.harvard.edu (P.C.); mlotz@mah.harvard.edu (M.L.); katherine.a.stafford@gmail.com (K.S.); abby.dangelo@mah.org (A.D.); plam1@mah.harvard.edu (P.L.); 2Department of Medicine, Harvard Medical School, Boston, MA 02129, USA; 3Sylvester Comprehensive Cancer Center, University of Miami, Miami, FL 33146, USA; 4Division of Hematology and Oncology, Mount Auburn Hospital, Cambridge, MA 02138, USA; 5Department of Pulmonary and Critical Care, Temple University, Philadelphia, PA 19122, USA; 36omaromary@gmail.com; 6Department of Pulmonary and Critical Care, Medical College of Wisconsin, Milwaukee, WI 53226, USA; drhpsingh101@gmail.com; 7Department of Pulmonary and Critical Care, University of Vermont, Burlington, VT 05405, USA; lipisha11@gmail.com; 8Department of Pulmonary and Critical Care, University of South Florida, Tampa, FL 33620, USA; rupal.arashdeep@gmail.com; 9Department of Medicine, Lahey Medical Center, Burlington, MA 01805, USA; qadir.d.9@gmail.com

**Keywords:** genetic screening, variant of uncertain significance (VUS), HBOC, Lynch syndrome

## Abstract

**Simple Summary:**

The increasing advancements in genetic testing have led to a rise in the number of patients undergoing cancer genetic testing who receive a VUS (variant of uncertain significance) result. This outcome leaves both patients and their healthcare providers perplexed, as they are unsure about the actual cancer risk and the necessary preventive measures. To address this issue, our retrospective study aimed to assess the occurrence of VUSs in patients tested for two prevalent cancer genetic syndromes. Additionally, we sought to explore the demographic and clinical characteristics of the population who received a VUS result. Our findings revealed that nearly one third of patients tested for common cancer genetic syndromes obtained a VUS test result. Furthermore, we discovered that age, personal history of breast cancer, and family history of breast or ovarian cancer were associated with VUS results. Further research is imperative to identify individuals at risk of receiving a VUS report and, more importantly, to develop tests that can accurately determine the associated cancer risk.

**Abstract:**

Hereditary Breast and Ovarian Cancer (HBOC) and Lynch Syndrome (LS) are the most common inherited cancer syndromes identified with genetic testing. Testing, though, commonly reveals variants of uncertain significance (VUSs). This is a retrospective observational study designed to determine the prevalence of pathogenic mutations and VUSs in patients tested for HBOC and/or LS and to explore the characteristics of the VUS population. Patients 18–80 years old that met NCCN criteria for HBOC and/or LS genetic screening were tested between 2006 and 2020 at Mount Auburn Hospital in Cambridge, Massachusetts. A total of 663 patients were included in the study, with a mean age of 50 years old and 90% being females. Pathogenic mutations were identified in 12.5% and VUSs in 28.3%. VUS prevalence was associated with race (*p*-value = 0.019), being particularly higher in Asian populations. Patients with a personal history of breast cancer or family history of breast or ovarian cancer were more likely to have a VUS (personal breast: OR: 1.55; CI: 1.08–2.25; family breast: OR: 1.68; CI: 1.08–2.60, family ovarian OR: 2.29; CI: 1.04–5.45). In conclusion, VUSs appear to be detected in almost one third patients tested for cancer genetic syndromes, and thus future work is warranted to determine their significance in cancer development.

## 1. Introduction

Germline genetic testing is slowly becoming a regular part of clinical practice. With the surge of next-generation sequencing technologies, many novel genetic variants have been identified, most of which are variants of uncertain significance (VUSs) [1,2]. The latter are DNA changes in a genetic sequence with an unknown effect on the gene function and risk for cancer development [3]. Thus, genetic tests for inherited mutations are reported as positive (pathogenic mutations likely to cause malignancy), negative (no mutation), or VUS (variation in a genetic sequence for which the association with disease risk is unclear) [4]. The two most common hereditary cancer syndromes are Hereditary Breast and Ovarian Cancer Syndrome (HBOC) and Lynch syndrome [5]. HBOC is responsible for 10–15% of all breast cancers and 24% of epithelial ovarian cancers [6,7]. Tumor suppressor genes BRCA1 and BRCA2 are most commonly involved in HBOC [6]. Lynch syndrome is an autosomal dominant disorder characterized by germline mutations, leading to impaired function in one of the multiple mismatch repair genes. As a result of impaired DNA processing correction, patients with Lynch syndrome are at 80% risk for colorectal cancer, 60% for endometrial cancer, and at increased risk for several other malignancies [8,9]. The mismatch repair genes commonly involved are *MLH1*, *MSH2*, *MSH6*, and *PMS2* [8,9].

As our understanding of hereditary cancer syndromes advances, a noteworthy convergence has emerged among these syndromes. This intersection is particularly evident in the increased risks associated with certain cancers. Both pancreatic and ovarian cancers, for instance, exhibit heightened risks for hereditary cancer syndromes [10,11,12,13]. Lynch syndrome (LS) alone accounts for 13–15% of ovarian cancer cases [14]. The literature also underscores the association between specific mutations and an elevated risk of certain cancers. BRCA1 and BRCA2 mutations, for instance, are linked to increased risks of endometrial and colorectal cancers [15], while Lynch syndrome is associated with a higher breast cancer risk [16]. Notably, Lynch syndrome patients with specific mutations, such as PMS2 and those exhibiting MMR deficiency, face an elevated risk of developing breast cancer [17,18]. The accumulating evidence designates breast cancer as an extra colonic manifestation of Lynch syndrome [17,18]. Furthermore, the co-occurrence of pathogenic variants in both Hereditary Breast and Ovarian Cancer Syndrome (HBOC) and Lynch syndrome is on the rise, underscoring the complex interplay between these genetic predispositions [19].

Genetic testing of high-risk individuals for HBOC and Lynch syndrome allows patients and their children to undergo close surveillance with more frequent screening, including breast imaging or colonoscopies, endometrial sampling, and pelvic ultrasounds with CA-125 levels, leading to early detection and improved morbidity and mortality [20,21]. Depending on the results, some may be placed on risk-reducing medications or decide to undergo risk-reducing surgery, like mastectomy and salpingo-oophorectomy [21,22]. Unfortunately, as NGS (Neo-Genomic Sequencing) use is becoming more widely available, an increasing proportion of patients tested for HBOC or Lynch receive a diagnosis of VUS, leaving them and their physicians unsure of how to proceed with screening intensity [1,23]. The reports of VUS rates vary, ranging from 10–40% in patients undergoing genetic testing for hereditary cancer syndromes [1,24,25]. Specific patient demographics or risk factors associated with a VUS result continue to remain unclear. The aim of our study is to determine the prevalence of pathogenic mutations and VUS in patients tested for HBOC and/or Lynch syndrome and to further explore the demographic and clinical characteristics of the population receiving a VUS result.

## 2. Materials and Methods

### 2.1. Study Design 

This is retrospective observational study designed to determine the incidence of genetic mutations and VUS in patients who underwent genetic risk assessment for HBOC and/or Lynch syndrome at Mount Auburn Hospital (MAH), an academic community hospital in Boston, Massachusetts, between 2006 and 31 July 2020. Patient demographics, clinical characteristics, and follow-up were gathered through electronic medical records. For data storage, a master code spreadsheet was generated linking a study number to direct identifiers (patient names, dates of birth, dates of therapy, dates of access to medical records, and medical record numbers) Then, a de-identified data spreadsheet was created with each patient’s study number, race, ethnicity, medical history, treatment history, family history, disease characteristics, and outcomes. Given this was a medical record review study without interventions, no informed consent was required. With regards to confidentiality, all spreadsheets were maintained as password-protection-accessible only to members of the study protocol, which was approved by the Institutional Review Board of our network. 

### 2.2. Study Population 

Included in the study were patients between 18 and 80 years who met the criteria for HBOC or Lynch syndrome genetic testing between 1 January 2006 and 31 July 2020. For HBOC, we used the NCCN recognized criteria for genetic testing as summarized in Table 1(a) [26]. For Lynch syndrome, genetic testing was pursued in individuals that met Amsterdam II or Bethesda criteria, as described in Table 1(b) [27,28]. 

### 2.3. Genetic Testing 

Genetic mutation information was extracted from patients’ genetic testing laboratory reports. All our patients underwent testing via the commercial Invitae or Myriad genetic panels. These panels encompass a comprehensive 20+ gene hereditary cancer panel, inclusive of both HBOC and Lynch genes, along with several others. The 20+ common genes for which patients are typically, at minimum, tested for are as follows: BRCA1, BRCA2, MLH1, MSH2, MSH6, PMS2, EPCAM, APC, MUTYH • CDK4, CDKN2A (p16INK4a, p14ARF), TP53, PTEN, STK11, CDH1, BMPR1A, SMAD4, PALB2, ATM, CHEK2, NBN, BARD1, BRIP1, RAD51C, RAD51D, POLD1, POLE, and GREM1. Patient records were collected for the period spanning from 2006 to 2020. Notably, genetic panels underwent modifications over time, resulting in variations among tests. These changes were influenced by the addition of new genes or, in some instances, patients opting for more limited panels based on their preferences or insurance coverage. Our analysis considered all Variants of Uncertain Significance (VUSs) detected through genetic testing, extending beyond the confines of NCCN-designated HBOC and Lynch syndrome genes. For result interpretations, reports without identified mutations were categorized as “negative”. Conversely, if a mutation was identified, it was classified as “positive” (indicating a pathogenic mutation), “VUS”, or grouped with the negative results if deemed benign.

To classify mutations, we utilized ClinVar 19, designating results as “positive” if they had at least one pathogenic report in ClinVar. Variants were considered VUS if they lacked pathogenic reports in ClinVar but had at least one report of unclear significance. Variants were designated as negative if they lacked pathogenic or unclear significance reports in ClinVar. The determination of results as positive, negative, or VUS was based on ClinVar reports published as of 9 July 2023. Mutations not previously reported in ClinVar were considered VUS and were included in our analysis.

### 2.4. Outcomes

Our primary goal was to determine the prevalence of pathogenic mutations and VUS in patients that underwent genetic risk assessment for HBOC and/or Lynch syndrome. We also assessed the demographic and clinical characteristics of the population that underwent genetic testing and tried to explore potential risk factors that could be associated with the detection of VUSs. 

### 2.5. Statistical Analysis 

The descriptive data for demographic and clinic characteristics are presented as median ± standard deviation (SD) or percentages (%). The Chi-square test was used to evaluate VUS proportion within different subgroups. A *p*-value ≤ 0.05 was considered significant. The Chi-square test was performed with GraphPad Prism, version 9.5.1. Effect size was measured with Odds Ratio (OR) for retrospective data, and the Confidence Interval (CI) was set at 95%. 

## 3. Results

There were 663 individuals included in our study. All patients met the criteria for genetic testing for HBOC or Lynch syndrome (or both) and were tested during the period 2006–2020 at MAH. The mean age was 50 years (SD:15), with 90% being females and with a high proportion of patients of Ashkenazi Jewish descent (14.3%). Regarding personal history of cancer, 162 (24.4%) had a previous breast cancer and 25 (3.7%) had a precancerous breast lesion. Colon cancer rates were much lower, at 3.01%. The majority of patients had a family history of cancer, with 351 (52.9%) in first- and 396 (59.7%) in second-degree relatives. In our cohort, 20 (3.01%) patients reported active smoking and 47 (7.08%) reported previous smoking history. Almost no patients reported heavy alcohol drinking (i.e., >7 drinks/week for females and >14 drinks/week for males). Baseline characteristics are available in Table 2. 

The most common indication for genetic testing was HBOC (558, 84.2%). A total of 90 pathogenic mutations were identified in 83 (12.5%) patients, with the most common being BRCA mutations (24, 28.9%), followed by MUTYH (11,13.2%) and CHEK2 mutations (10, 12.0%). All pathogenic mutations were reported classified as pathogenic in ClinVar, and among them, 12 out of 90 (13.3%) had conflicting reports indicating uncertain significance.

Among the whole group, 188 (28.3%) patients had a VUS, with 253 total VUSs identified (Table 3). Eleven of those mutations had not been previously reported in ClinVar (Table 4). Fifty-two patients (26.6%) had two or more VUSs within the same or different genes. Most VUSs were in APC, followed by ATM and MSH3 (Figure 1). Other common genes in which VUSs were detected were CHEK2, NBN, BRIP1, MSH2, and MSH6 (Figure 1). ATM and MSH6 were the most common genes in the Asian population. In the African-American population, the most common genes were NBN and APC. All variants found were previously described in ClinVar. VUSs and pathogenic mutations were present simultaneously in nine patients (0.90%). 

During the period from June 2021 to June 2023, we observed significant changes in the distribution of pathogenic, VUS, and negative findings within our cohort, driven by the updates to ClinVar. Notably, our preliminary analysis of 663 patients in June 2021 revealed 76 deleterious mutations, whereas the June 2023 analysis identified 89 such mutations. Additionally, the number of VUSs also saw an upward trend, rising from 200 in June 2021 to 253 by June 2023. We also identified VUSs that had conflicting data on ClinVar (Table 5).

Rates of VUS were not different with relation to age (≤40 vs. >40) or ethnicity (Hispanic vs Non-Hispanic) (Table 6). Interestingly, VUS incidence rates were significantly associated with race (Table 6). Post hoc analysis showed that VUSs were much more commonly detected in our Asian population compared to White patients (OR: 2.44; CI: 1.18–5.15). There was no difference between White and African-American, or African-American vs Asian. Furthermore, patients with a personal history of breast cancer were almost two times more likely to have a VUS compared to patients without a previous history (OR: 1.55; CI: 1.08–2.25) (Table 6). Patients with a family history of breast or ovarian cancer in a first-degree relative were also more likely to have a VUS (OR: 1.68; CI: 1.08–2.60 for breast and OR: 2.29; CI: 1.04–5.45 for ovarian) (Table 6). Family history of prostate or colon cancer in a close relative did not affect VUS rates (Table 6). Finally, the most common recommendation following a VUS report was annual breast magnetic resonance imaging (MRI) (70/188, 37.2%) as per review of clinical notes on electronic medical records. The next most common recommendations were following up with the primary care doctor (PCP) (20/188, 10.6%) or oncologist (30/188, 15.9%) in high-risk clinics.

## 4. Discussion 

To our knowledge, this is the first study evaluating the prevalence of VUS in patients meeting genetic test criteria for HBOC or Lynch syndrome in a USA-based population between 2006 and 2020. Our results show that 12.5% of the 663 patients that met the criteria for HBOC and/or Lynch genetic testing were found to have a pathogenic mutation, while almost one third of patients (28.3%) were found to have a VUS. VUS detection appeared to be associated with race, personal history of breast cancer, and family history of breast or ovarian cancer. 

In our patient population meeting genetic test criteria for HBOC or Lynch syndrome, 12.5% of patients were found to have a pathogenic mutation, with the most common being in BRCA1/2 followed by MUTYH and CHEK2. BRCA1/2 are the most common genes involved in HBOC, and mutations in either of the *BRCA* genes increase a woman’s risk of breast cancer to 45–65% by 70 years old [29]. CHEK2 DNA repair gene pathogenic mutations also account for a significant amount of breast and colon cancer [30]. In contrast, MUTYH is primarily associated with MUTYH-associated polyposis, exhibiting some phenotypic similarities to Lynch syndrome [31]. In a Europe-based study, MUTYH accounted for only 3.6% of Lynch syndrome–like cases. In our study, however, we identified MUTYH pathogenic mutations in 13.2% of cases. This disparity may be attributed to incidental factors, population-specific genetic variability, or other contributing factors.

The main finding of our manuscript is the high prevalence of VUSs in patients tested for HBOC and/or Lynch. Almost one-third (28.3%) of the patients received a report for a VUS. There have only been a few studies evaluating VUS prevalence across the world, generating inconsistent results. For example, VUS prevalence was 46.1% in a Brazilian population tested for HBOC, while it was 9.2% in a Jordanian-Arab population [32,33]. This variability can be attributed to the availability of genetic testing, the number of genes tested in each panel, and possibly demographics and specific population characteristics. It is also important to consider that the relevance of genetic variants to disease is undergoing constant evaluation and change as NGS continues to advance. This is evident from the significant difference in prevalence numbers observed between our June 2021 and June 2023 analyses. The shifting prevalence of numbers highlights the continuous discovery of new variants and the evolving landscape of genetic research. Staying updated on the latest advancements is essential to ensure accurate and up-to-date genetic analysis and interpretation.

In addition, VUS prevalence showed no significant difference between the Hispanic and non-Hispanic populations in our study. However, when examining different racial groups, we observed significantly higher VUS rates in all non-White groups. A subgroup analysis further revealed that a significant difference in VUS rates was specifically observed between the Asian and White populations. However, the small sample size in the non-White groups and uneven distribution due to the observational study design may pose limitations to our findings. The literature data regarding the prevalence of VUSs based on race or ethnicity are currently scarce. Some studies find no correlation, while others, consistent with our results, find a higher prevalence of VUSs in non-White populations [34,35]. In a recent large study, Ndugga-Kabuye et al. evaluated VUS prevalence in 50,000 patients and confirmed the higher prevalence of VUS in non-European populations (i.e., Hispanics, African Americans, Asians, and Pacific Islanders). The absence of ancestral variety in genomic investigations is most likely to blame for the racial/ethnic discrepancies. For example, European ancestral populations account for the majority of observations (55.8%) in ClinVar, used to determine if a mutation is pathogenic or of undetermined significance. 

Furthermore, we show that individuals with a personal history of breast cancer or family history of breast/ovarian cancer are more likely to have a VUS compared to their counterparts without history. This association may be an unintended consequence of VUS predominance in our population that will be later reclassified as pathogenic. If not incidental, it raises the question of whether VUS is the outcome of the increased mutational tendency seen in patients with pathogenic mutations and/or a history of cancer. To the best of our knowledge, no previous studies have compared the prevalence of VUS in patients with and without a personal or family history of cancer. Ndugga-Kabuye et al. performed separate analyses for VUS prevalence for patients with a positive and negative history of cancer. Although they did not statistically compare the prevalence, based on descriptive data, a VUS in HBOC or LS genes was found in 6.8% of patients with a personal history of cancer and in 6.4% of patients without, which is more suggestive of a similar distribution [35]. Another study that looked at VUS prevalence in breast cancer found no link between VUS rates and triple-negative malignancy [33]. Larger, prospective, and randomized studies are thus needed to investigate the possible correlations of VUS with ancestry and cancer history. 

The unclear diagnosis resulting from a VUS result can have a negative impact on a patient’s health. The lack of structured medical guidance leads to inconsistent management based on the approach of the PCP or oncologists [3]. Some patients may not be offered predictive testing, while others may enter a cycle of intense screening and/or risk-reducing interventions of unclear benefit [3]. Furthermore, studies show that an inconclusive diagnosis may affect patients’ psychology, increasing stress and anxiety [3,36]. Similarly, the accuracy of pathogenic reports may also need to be questioned. In our study, all pathogenic mutations were listed as pathogenic on ClinVar, but 12/90 (13.3%) also had conflicting reports for uncertain significance or benign classification. This can lead to overestimation of the pathogenicity of the assessed variants and underestimation of VUSs. Thus, there is an unmet need to understand and better classify VUSs. 

Towards this end, the ongoing development of functional assays holds promise in accurately determining the cancer risk associated with VUSs in a timely way [37,38]. These assays are designed to assess the impact of a particular variant on the structure and functionality of the corresponding protein, providing an estimated VUS pathogenicity [37]. Large investigator consortia, like ENIGMA or INSIGHT, have started to form, with the aim of collecting a large amount of data on VUSs and subsequently using machine learning prediction models or functional assays in determining the significance of variants [39,40,41]. 

To further address the complexity of sequence interpretation, the American Society of Medical Genetics and Genomics (ACMG) has released a set of guidelines for the classification of new sequences [4]. According to the ACMG, sequences should be categorized as pathogenic, likely pathogenic, of uncertain significance, likely benign, or benign. The criteria for classifying a sequence as pathogenic or likely pathogenic take various factors into consideration, such as the type of mutation (e.g., frameshift, nonsense), whether the mutation is causing an amino acid change of known pathogenicity, and the patient’s demographics. On the other hand, the criteria for benign or likely benign involve considerations like patient demographics, the frequency of the sequence, and existing functional studies. It is important to highlight that both sets of criteria take into great consideration the sequence distribution in healthy vs. affected individuals. Sequences identified in patients with a personal history provide robust evidence of pathogenicity, while those observed in healthy adults are more likely to be benign. Of note, these guidelines are designed for the interpretation of sequences related to inherited (Mendelian diseases) and not somatic genetic variations. 

For sequence interpretation, the ACMG recommends that healthcare providers work in collaboration with clinical laboratories, as the patient’s clinical information plays a crucial role in enabling the laboratory to accurately classify genetic variants [4]. Given the growing complexity of analysis and interpretation, the ACMG strongly advocates for these tests to be conducted in Clinical Laboratory Improvement Amendments (CLIA)-approved laboratories, and the results should be interpreted by a board-certified clinical molecular geneticist or pathologist [4]. While acknowledging the current imperfections in variant classification, the ACMG recommends that mutations deemed pathogenic or likely pathogenic have substantial evidence to be used in clinical decision making, always within the appropriate clinical context [4]. On the other hand, variants of uncertain significance (VUSs) are discouraged from influencing clinical decisions [4]. Some experts recommend that laboratories do not release VUS reports and alert the clinician only if the VUS is reclassified as pathogenic [42]. However, this approach might threaten patients’ autonomy. As per the ACMG, as efforts to reclassify VUS are ongoing, vigilant monitoring of patients with VUS results is advisable [4]. This approach ensures a cautious and informed approach to utilizing genetic information in clinical practice.

## 5. Conclusions

In conclusion, VUSs appear to be detected in almost one out of three patients tested for cancer genetic syndromes, like HBOC and Lynch, thus having a huge impact on patients’ psychology and health management. Further studies are required to identify patients at risk for VUSs and, most importantly, to develop tests to determine the associated cancer risk. 

## Figures and Tables

**Figure 1 cancers-15-05762-f001:**
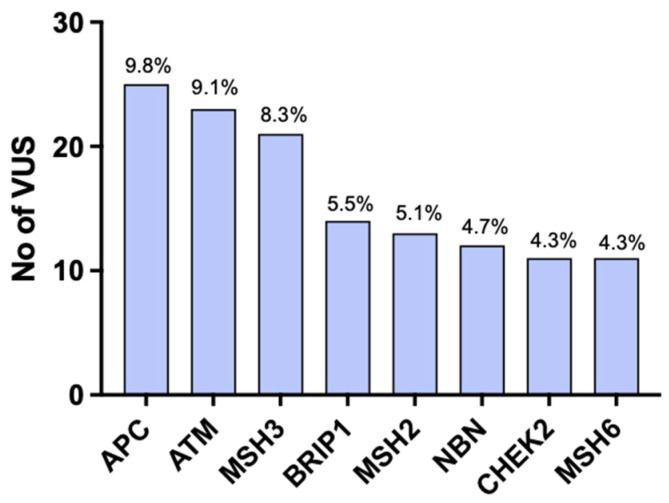
Most common genes in which VUSs were detected.

**Table 1 cancers-15-05762-t001:** (**a**) Indications for HBOC genetic testing. (**b**) Indications for Lynch syndrome genetic testing.

(**a**)	
Personal history of cancer	Breast cancer diagnosed at ≤50 y.o.
	Triple-negative breast cancer
	≥2 primary breast cancers
	Lobular breast cancer with concurrent history of diffuse gastric cancer
	Epithelial ovarian cancer, fallopian cancer, or primary peritoneal cancer
	Male breast cancer
	Ashkenazi Jewish ancestry
	High-risk family history:
	≥1 close relative with breast cancer ≤ 50, male breast cancer, ovarian cancer, pancreatic cancer, or metastatic or high-risk group prostate cancer;
	≥3 breast cancer diagnoses in patient and/or close relatives;
	≥2 close relatives with breast or prostate cancer.
No personal history of cancer	Close relative meeting any of the above criteria
	Individual with >5% of BRCA1/2 pathogenic variant in risk calculators (i.e., Tyrer-Cuzick, BRCAPro, CanRisk)
(**b**)	
Amsterdam II	Three relatives with Lynch syndrome–related cancers with all the following criteria met:
	One is a first-degree relative of the other two;
	Lynch syndrome–related cancer affecting more than one generation;
	At least one Lynch syndrome–related cancer diagnosed at ≤50 y.o.;
	Familial adenomatous polyposis should be excluded;
	Tumors should be verified by pathologic examination.
Bethesda Criteria	Test if any of the following:
	CRC diagnosed at ≤50 y.o.;
	Presence of synchronous or metachronous Lynch syndrome–related tumors;
	CRC with MSI-high histology diagnosed at ≤50 y.o.;
	CRC diagnosed in a patient with ≥1 first-degree relatives with a Lynch syndrome–related cancer, with one of the cancers diagnosed at ≤50 y.o.;
	CRC diagnosed in a patient with ≥2 first- or second-degree relatives with Lynch syndrome–related cancer.

Close relative refers to first-, second-, or third-degree relative on the same side of the family. Lynch syndrome–related cancers: colorectal, endometrial, gastric, ovarian, pancreatic, urothelial, brain (usually glioblastoma), biliary tract, and small intestine sebaceous adenomas, sebaceous carcinomas, and keratoacanthomas. CRC: Colorectal cancer. MSI: Microsatellite instability.

**Table 2 cancers-15-05762-t002:** Patient characteristics (n = 663).

	Number of Total (%)	
**Age (mean ± SD)**	50 ± 15	
**Gender**		
Males	63 (9.50%)	
Females	597 (90.04%)	
Transgender	3 (0.45%)	
**Ethnicity**		
Hispanic	35 (5.27%)	F: 32/35
Non-Hispanic	378 (57.01%)	F: 338/378
Unknown	250 (37.70%)	F: 227/250
**Race ***		
White	333 (50.20%)	F: 298/333
African American	20 (3.01%)	F:17/20
Asian	34 (5.12%)	F: 30/34
Native American	10 (1.50%)	F: 10/10
Unknown	280 (42.23%)	F: 254/280
**Ashkenazi Jews**	95 (14.3%)	
**Personal history of breast cancer**	162 (24.43%)	
**Personal history of colon cancer**	20 (3.01%)	
**Family history of cancer in 1st-degree relative**	351 (52.94%)	
Breast cancer in 1st-degree relative	116 (17.49%)	
Ovarian cancer in 1st-degree relative	23 (3.46%)	
Prostate cancer in 1st-degree relative	31 (4.67%)	
Colon cancer in 1st-degree relative	21 (3.16%)	
**Family history of cancer in 2nd-degree relative**	396 (59.7%)	

* Some patient were of biracial descent and thus are counted in more than one racial group. F: Refers to the proportion of females within this particular ethnicity or race group.

**Table 3 cancers-15-05762-t003:** VUSs in patients tested for HBOC and/or Lynch syndrome.

Age at Testing	Syndrome Tested for	VUS	Reference Sequence (RS)
51	HBOC	c.1190A>G (p.Gln397Arg) in AXIN2 c.3650C>G (p.Ser1217Cys) in BRCA1 c.1027-1G>T (splice acceptor) in RAD51C	rs774887154 rs398122676 rs1567818502
52	HBOC	c.8246A>T (p.Lys2749Ile) in ATM c.556A>G (p.Ser186Gly) in BARD1 c.743G>A (p.Arg248Gln) in MSH6	rs779145081 rs16852741 rs764870249
71	HBOC	c.803C>T (p.Pro268Leu) in RECQL4 c.899A>G (p.Gln300Arg) in SMARCE1	rs760340046 rs766568737
37	HBOC	c.4002-8dup in MSH6 c.210A>G (p.Ser70Ser) in PALB2	rs267608139 rs786202650
24	HBOC	c.6179G>A (p.Arg2060His) in ATM c.470T>C (p.Ile157Thr) in CHEK2	rs376521407 rs17879961
79	HBOC	c.3440dup (p.Asn1147fs) in BRIP1 c.470T>C (p.Ile157Thr) in CHEK2	rs753683450 rs17879961
35	HBOC	c.2075A>C (p.His692Pro) in BRCA1 c.730A>G (p.Ile244Val) in RAD51C	rs2053831947 rs199886026
42	HBOC	c.1348A>C (p.Asn450His) in PALB2 c.3362G>A (p.Gly1121Asp) in PALB2	rs62625274 rs62625282
62	HBOC	c.6865A>G (p.Thr2289Ala) in APC c.1655T>C (p.Ile552Thr) in BRIP1	rs1554087807 rs369340666
50	HBOC	c.7415A>C (p.Lys2472Thr) in BRCA2 c.5C>T (p.Ser2Phe) in MSH3	rs80358963 rs768844493
82	HBOC	c.722C>T (p.Ala241Val) in STK11	rs2080777192
50	HBOC	c.970G>A (p.Glu324Lys) in SDHA	rs147014102
35	HBOC	c.26G>C (p.Cys9Ser) in RAD51D	rs140825795
46	HBOC	c.961G>A (p.Gly321Ser) in POLD1	rs41554817
70	HBOC	c.1168C>T (p.Pro390Ser) in POLD1	rs2038747136
73	HBOC	c.1425G>T(p.Glu475Asp) in PDGFRA	rs200309940
42	HBOC	c.1186G>T (p.Ala396Ser) in PDGFRA	rs1327567130
35	HBOC	c.2317A>G (p.Met773Val) in PDGFRA	rs191808397
65	HBOC	c.3014T>C (p.Phe1005Ser) in PALB2	rs879254268
60	HBOC	c.3025C>G (p.Pro1009Ala) in PALB2	rs764669864
64	HBOC	c.821T>C (p.Leu274Pro) in NTHL1	Variation ID: 2517325
69	HBOC	c.458G>A (p.Arg153Gln) in NTHL1	Variation ID: 2518789
60	HBOC	c.1720T>A (p.Leu574Ile) in NBN	rs142334798
74	HBOC	c.643C>T (p.Arg215Trp) in NBN	rs34767364
31	HBOC	VUS in NBN (Missing specific sequence)	NA
81	HBOC	VUS in NBN (Missing specific sequence)	NA
63	HBOC	c.2060A>C (p.Lys687Thr) in NBN	rs186371605
67	HBOC	c.1447A>G (p.Thr483Ala) in MUTYH	Not previously reported
60	HBOC	c.1306C>G (p.Leu43Val) in MUTYH	Not previously reported
68	HBOC	c.1778G>A (p.Arg593Gln) in MSH3	rs764832633
37	HBOC	c.1764-9_1764-8del in MSH3	rs41559616
79	HBOC	VUS in MSH3 (Missing specific sequence)	NA
41	HBOC	c.2732T>G (p.Leu911Trp) in MSH3	rs41545019
71	HBOC	c.1028-6T>C in MSH3	rs769258876
48	HBOC	VUS in MSH2 (Missing specific sequence)	NA
49	HBOC	VUS in MLH1 (Missing specific sequence)	NA
67	HBOC	c.359G>C (p.Arg120Pro) in GALNT12	rs202137559
64	HBOC	c.907G>A (p.Asp303Asn) in GALNT12	rs145236923
75	HBOC	c.4010A>C (p.Asp1337Ala) in DICER1	rs1891070773
50	HBOC	c.248A>G (p.Tyr83Cys) in DICER1	rs373646414
61	HBOC	VUS in CTNNA1 (Missing specific sequence)	NA
44	HBOC	c.1111C>T (p.His371Tyr) in CHEK2 c.2357T>C (p.Val786Ala) in MSH3	rs531398630
32	HBOC	c.1111C>T (p.His371Tyr) in CHEK2 c.719C>T (p.Pro240Leu) in GALNT12	rs531398630 rs59362219
58	HBOC	c.580A>T (p.Ser194Cys) in CHEK2	rs786203042
53	HBOC	c.593-3_593-2insSVA in CHEK2	Not previously reported
46	HBOC	c.369T>A (p.His123Gln) in CHEK2	Not previously reported
55	HBOC	c.-2G>A in CDKN2A	rs191394143
69	HBOC	c.310C>A (p.Leu104Met) in CDK4	rs759535768
61	HBOC	c.1897A>C (p.Ile663Leu) in BRIP1	rs765314472
52	HBOC	c.2469G>T (p.Arg823Ser) in BRIP1	rs587780239
31	HBOC	c.415T>G (p.Ser139Ala) in BRIP1	rs202072866
47	HBOC	c.226G>A (p.Val76Ile) in BRIP1	rs769573395
40	HBOC	c.2423G>T (p.Arg808Ile) in BRIP1	rs781153382
75	HBOC	c.9816T>G (p.Asp3272Glu) in BRCA2	rs56111359
32	HBOC	c.2779A>G (p.Met927Val) in BRCA2	rs786201837
36	HBOC	R2784Q (8579G>A) in BRCA2	rs80359076
33	HBOC	c.3318C>G (p.Ser1106Arg) in BRCA2	rs1298550035
48	HBOC	c.1360C>G (p.Pro454Ala) in BARD1	rs730881408
76	HBOC	c.2284T>C (p.Trp762Arg) in BARD1	rs878854008
78	HBOC	VUS in AXIN2 (Missing specific sequence)	NA
32	HBOC	c.1235A>G (p.Asn412Ser) in AXIN2	rs115931022
67	HBOC	c.1985T>C (p.Leu662Pro) in AXIN2	rs142476324
79	HBOC	c.6019C>T (p.Leu762Arg) in ATM	Not previously reported
43	HBOC	c.7871G>C (p.Cys2624Ser) in ATM	rs759392666
51	HBOC	c.2698A>G (p.Met900Val) in ATM	rs138468963
35	HBOC	VUS in ATM (Missing specific sequence)	NA
48	HBOC	c.6919C>T (p.Leu2307Phe) in ATM	rs56009889
39	HBOC	c.8155C>T (p.Arg2719Cys) in ATM	rs138526014
74	HBOC	VUS in ATM (Missing specific sequence)	NA
39	HBOC	VUS in ATM (Missing specific sequence)	NA
48	HBOC	c.2011A>G (p.Ile671Val) in ATM	rs730881344
57	HBOC	c.6248T>C (p.Ile2083Thr) in APC	rs758715972
48	HBOC	VUS (Missing specific sequence)	NA
57	HBOC	c.1169C>T (p.Thr390Met) in TSC2	rs1596303442
33	HBOC	c.758A>G (p.His253Arg) in SMARCA4	rs2086063382
47	HBOC	c.367C>T (p.Pro123Ser) in SDHC c.1307C>T (p.Ala436Val) in TERT	rs773039986 rs986886145
33	HBOC	c.172A>T (p.Thr58Ser) in RNF43	rs142864107
57	HBOC	c.1720A>C (p.Lys574Gln) in RAD50	rs779597467
40	HBOC	c.6767G>C (p.Gly2256Ala) in POLE	rs749707316
60	HBOC	c.101G>T (p.Arg34Leu) in POLE	rs747005851
38	HBOC	c.1016A>T (p.Asp339Val) in POLE	rs1060500865
38	HBOC	c.75del (p.Asp25Glufs*16) in POLD1	rs772855121
57	HBOC	c.1510G>C (p.Glu504Gln) in PMS2	rs368516768
33	HBOC	c.97A>T (p.Asn33Tyr) in PDGFRA	rs200979664
32	HBOC	c.2317A>G (p.Met773Val) in PDGFRA	rs191808397
72	HBOC	c.1730C>T (p.Pro577Leu) in PDGFRA	rs778015444
63	HBOC	c.1651C>A (p.Gln551Lys) in PDGFRA	rs770950644
53	HBOC	c.155T>C (p.Val52Ala) in PALB2	rs373970237
41	HBOC	c.208G>A (p.Gly70Ser) in NTHL1	Variation ID: 2096931
41	HBOC	c.5360C>T (p.Thr1787Met) in NF1	rs760649828
27	HBOC	c.4009C>T (p.Arg1337Trp) in NF1	rs146306756
49	HBOC	c.3883A>G (p.Thr1295Ala) in NF1	rs143836226
27	HBOC	c.3315A>G (Silent) in NF1	rs1555614915
51	HBOC	c.169G>A (p.Gly57Ser) in NF1	rs779727341
48	HBOC	c.595C>T (p.Pro199Ser) in NBN	rs587780097
61	HBOC	c.536A>T (p.Glu179Val) in NBN	rs864622578
47	HBOC	c.430A>G (p.Thr144Ala) in NBN	rs1812023859
56	HBOC	c.2056A>G (p.Lys686Glu) in NBN	rs786203920
44	HBOC	c.1343A>T (p.Gln448Leu) in NBN c.1120A>G (p.Ser374Gly) in TSC1	rs146403088 Variation ID: 1063383
43	HBOC	c.100G>C (p.Glu34Gln) in MUTYH	rs1557492431
42	HBOC	c.458G>C (p.Gly153Ala) in MSH6 c.3290C>T (p.Pro1097Leu) in PALB2	rs1251899870 rs587781308
39	HBOC	c.1720C>T (p.Arg574Trp) in MSH3 c.3762A>T (p.Glu1254Asp) in MSH6	rs771054581 rs375459388
42	HBOC	c.909G>C (p.Lys303Asn) in MSH3	rs757164724
82	HBOC	c.582C>G (p.Asp194Glu) in MSH3	rs749446559
37	HBOC	c.350G>A (p.Gly117Asp) in MSH3	rs1456712758
31	HBOC	c.2732T>G (p.Leu911Trp) in MSH3	rs41545019
56	HBOC	c.2185C>G (p.His729Asp) in MSH3	rs145353158
71	HBOC	c.2173G>A (p.Glu725Lys) in MSH3 c.845C>T (p.Thr282Ile) in MSH3 c.1255G>A (p.Ala419Thr) in MUTYH	rs200612739 rs202184623 rs58778044
52	HBOC	c.1568+5G>A (Intronic) in MSH3	rs778804919
61	HBOC	c.1019T>C (p.Ile340Thr) in MSH3	rs1228031532
64	HBOC	c.1172C>A (p.Ala391Asp) in MSH2	rs864622674
59	HBOC	c.1064G>A (p.Arg355Lys) in MSH2	Variation ID: 1781308
66	HBOC	c.1856A>G (p.Tyr619Lys) in MSH2	rs63749982
28	HBOC	c.1172C>A (p.Ala391Asp) in MSH2	rs864622674
84	HBOC	c.2606C>A (p.Ala869Glu) in MSH2 c.257C>T (p.Ala86Val) in POLD1	rs730881772 rs148040399
73	HBOC	c.123C>G (p.Asp41Glu) in MSH2	rs761960690
43	HBOC	c.2156T>C (p.Ile719Thr) in MLH1	rs757603534
39	HBOC	c.2045T>C (p.Met682Thr) in MLH1 c.1408A>G (p.Thr470Ala) in PALB2	rs1060500693 rs150636811
76	HBOC	c.941G>A (p.Arg314Gln) in MEN1	rs771645621
44	HBOC	c.1553C>T (p.Pro518Leu) in KIT c.334A>G (p.Asn112Asp) in MSH6	rs569408054 rs864622397
77	HBOC	c.429C>G (p.Phe143Leu) in FLCN c.2377G>A (p.Gly793Ser) in PALB2	Variation ID: 388510 rs878855109
67	HBOC	c.4352G>C (p.Arg1451Thr) in DICER1	Variation ID: 1056894
48	HBOC	c.1143+5T>C (Intronic) in CTNNA1	rs766106863
25	HBOC	c.539G>T (p.Arg180His) in CHEK2	rs137853009
51	HBOC	c.772A>G (p.Ile258Val) in CHEK2 c.-2A>C in RAD51D	rs876658690 rs2091800072
61	HBOC	c.663C>G (p.Ile221Met) in CHEK2	rs200451612
36	HBOC	c.331G>T (p.Asp111Tyr) in CHEK2 c.1019T>C (p.Phe340Ser) in MSH6 c.6674G>A (p.Arg2225His) in POLE	rs1569159072 rs61753793 rs538875477
39	HBOC	c.1567C>G (p.Arg523Gly) in CHEK2 c.4859C>T (p.Ser1620Phe) in SMARCA4	rs149501505 rs1600649021
41	HBOC	c.949T>C (p.Phe317Leu) in CDH1 c.2085A>G (Silent) in MSH3	rs1555515643 rs777245977
35	HBOC	c.1784C>G (p.Pro595Arg) CDH1	rs1555516843
39	HBOC	c.436A>G (p.Ile146Val) in BRIP1	rs1567868598
25	HBOC	c.3533A>T (p.Glu1178Val) in BRIP1 c.3379A>G (p.Asn1127Asp) in MSH3 c.1009G>A (p.Val337Ile) in PDGFRA	rs752850661
64	HBOC	c.337A>C (p.Thr113Pro) in BRIP1 c.728G>A (p.Arg243Gln) in MSH2	rs1555617812 rs63751455
53	HBOC	c.2233G>A (p.Ala745Thr) in BRIP1	rs587780235
63	HBOC	c.1660C>G (p.Gln554Glu) in BRIP1	rs777217004
56	HBOC	c.891_902del (p.Glu297_Val300del) in BRCA2	rs2072399471
36	HBOC	c.6703A>T (p.Met2235Leu) in BRCA2	Variation ID: 1056020
83	HBOC	c.343A>G (p.Lys115Glu) in BRCA2 c.1075C>A (p.Pro359Thr) in MUTYH	rs56242644 Not previously reported
33	HBOC	c.4339C>A (p.Gln1447Lys) in BRCA1	rs1567868598
50	HBOC	c.1022G>T (p.Gly341Val) in BMPR1A	rs1564724250
51	HBOC	c.80C>A in BARD1	NA
36	HBOC	c.748T>C (p.Ser250Pro) in BARD1 c.1618G>A (p.Asp540Asn) in MUTYH	rs570022823 Not previously reported
38	HBOC	c.617A>G (p.Gln206Arg) in BARD1	rs760718143
54	HBOC	c.1835A>T (p.Asp612Val) in BARD1	rs201140528
55	HBOC	c.2770C>T (p.Arg924Trp) in ATM	rs55723361
62	HBOC	c.8968G>A (p.Glu2990Lys) in ATM	rs1800558
37	HBOC	c.8187A>C (p.Gln2729His) in ATM c.821G>T (p.Gly274Val) in CDH1	rs587781946 rs876660861
51	HBOC	c.7743C>A (p.Ser2581Arg) in ATM	rs2086306575
54	HBOC	c.670A>G (p.Lys224Glu) in ATM c.1004A>G (p.Asn335Ser) in PMS2	rs145053092 rs200513014
58	HBOC	c.4375G>A (p.Gly1459Arg) in ATM	rs145667735
26	HBOC	c.4349T>C (p.Leu1450Pro) in ATM c.1192C>T (p.Arg398Cys) in GALNT12	rs750306932 rs747755624
39	HBOC	c.238C>T (p.Pro80Ser) in ATM	rs750597831
37	HBOC	c.133C>T (p.Arg45Trp) in ATM c.7457C>T (p.Thr2486Ile) in NF1	rs3218684 Not previously reported
36	HBOC	c.4088A>G (p.Lys1363Arg) in APC c.5392A>G (p.Asn1798Asp) in APC c.1489A>G (p.Ile497Val) in MSH2 c.157G>T (p.Ala53Ser) in MSH2	rs373607243 rs200794097 rs755501968 rs755931648
52	HBOC	c.6944A>G (p.Gln2315Arg) in APC c.1037C>T (p.Ser346Phe) in MSH6 c.503C>G (p.Ala168Gly) in MSH6	rs1060503273 rs567785169 rs774162322
79	HBOC	c.7903A>G (p.Thr2635Ala) in ATM	rs886059799
42	HBOC	c.2438A>G (p.Asn813Ser) in ATM	Not previously reported
38	HBOC	c.8462A>G (p.Asp2821Gly) in APC	rs780049836
68	HBOC	c.8276G>A (p.Arg2759His) in APC c.582C>G (p.Asp194Glu) in MSH3	rs538289470 rs749446559
41	HBOC	c.7399C>A (p.Pro2467Thr) in APC c.5026A>G (p.Arg1676Gly) in APC	rs372305287 rs370560998
29	HBOC	c.688C>T (p.Arg230Cys) in APC	rs587779805
45	HBOC	c.6724A>G (p.Ser2242Gly) in APC c.511A>G (p.Ile171Val) in NBN	rs201375478 rs61754966
53	HBOC	c.6520A>G (p.Ser2174Gly) in APC c.1117G>A (p.Gly373Arg) in MLH1	rs754536901 rs587776934
48	HBOC	c.6338G>C (p.Ser2113Thr) in APC	rs1766189874
28	HBOC	c.5240T>C (p.Met1747Thr) in APC c.1094G>A (p.Arg365Gln) in RAD50	rs864622751 rs146370443
23	HBOC	c.5216A>G (p.Lys1739Arg) in APC	rs769558291
66	HBOC	c.5026_5028del (p.Arg1676del) in APC c.3715A>G (p.Ile1239Val) in MSH6 c.668T>C (p.Ile223Thr) in RAD50	rs768369050 rs1469961964 rs1750475890
40	HBOC	c.4372C>T (p.Pro1458Ser) in APC	rs143796828
34	HBOC	c.2222A>G (p.Asn741Ser) in APC	rs150209825
62	HBOC	c.-30369A>G (Non-coding) in APC	NA
59	HBOC	c.203G>A, p.Arg68Gln in AXIN2	rs138056036
26	HBOC	c.1267C>T (p.Leu423Phe) in AXIN2 c.1567G>A (p.Glu523Lys) in MSH3	rs376630432 rs34058399
55	HBOC	c.111G>T (p.Gln37His) in AXIN2 c.1643G>A (p.Gly548Asp) in MSH2 c.1361G>A (p.Arg454Gln) in MSH3	Variation ID: 1494944 rs1573553753 rs144798521
79	HBOC	c.3352A>G (p.Asn1118Asp) in APC c.1660C>T (p.Arg554Trp) in RNF43	rs140493115 Variation ID: 1140674
47	HBOC	c.797C>G (p.Thr266Ser) in BMPR1A c.3762A>T (p.Glu1254Asp) in MSH6	rs1554890797 rs375459388
35	Lynch Syndrome	c.6363_6365dupTG (p.Ala2122dup) in APC c.2804C>T (p.Thr935Met) in ATM	rs587780602 rs3218708
48	HBOC	c.dup exon 2 (p14ARF) in CDKN2A c.dup entire (p16INK4a) in CDKN2A	Not previously Not previously reported
60	HBOC	c.5026A>G (p.Arg1676Gly) in APC c.7399C>A (p.Pro2467Thr) in APC	rs200794097 rs372305287
52	HBOC	c.626A>G (p.Ile2076Val) in ATM c.317G>C (p.Arg106Thr) in MSH2	Not previously reported rs41295286
47	HBOC	c.1007A>G (p.Asn336Ser) in POLE c.667C>T (p.Arg223Cys) in RNF43	rs5744760 rs755478993
73	HBOC	c.1353A>G (p.Asn118Ser) in BARD1 c.2081C>G (p.Pro694Arg) in NBN	rs142864491 rs746090959
50	HBOC	VUS in KIT and MSH6 (Missing specific sequence)	NA
70	HBOC	c.1655G>A (p.Arg552Lys) in GALNT12 c.527T>C (p.Ile176Thr) in NHTL1	rs1285871027 rs1805378
35	HBOC	c.3444C>A (p.Asp1148Glu) in BRIP1 c.-2G>A in CDKN2A	rs28997573 rs191394143
36	HBOC	VUS in ATM and RAD51D (Missing specific sequence)	NA
50	Lynch Syndrome	c.1883A>G (p.Asn628Ser) in DICER1	rs756051157
45	HBOC	c.536A>G (p.Tyr170Cys) in NHTL1	Not previously reported
43	HBOC	c.118G>A (p.Gly40Ser) in MSH2	rs63751260

For VUSs for which a reference sequence is not yet available, we provide the Variation ID from ClinVar (https://www.ncbi.nlm.nih.gov/clinvar/ (accessed on 26 November 2023)); HBOC: Hereditary Breast and Ovarian Cancer Syndrome; NA: reference sequence could not be provided as the full sequence was not available after data extraction.

**Table 4 cancers-15-05762-t004:** VUSs not previously reported.

Gene	VUS
ATM	c.6019C>T (p.Leu762Arg)
ATM	c.626A>G (p.Ile2076Val)
ATM	c.2438A>G (p.Asn813Ser)
CHEK2	c.593-3_593-2insSVA
CHEK2	c.369T>A (p.His123Gln)
MUTYH	c.1447A>G (p.Thr483Ala)
MUTYH	c.1306C>G (p.Leu43Val)
MUTYH	c.1075C>A (p.Pro359Thr)
MUTYH	c.1618G>A (p.Asp540Asn)
NF1	c.7457C>T (p.Thr2486Ile)
NHTL1	c.536A>G (p.Tyr170Cys)

**Table 5 cancers-15-05762-t005:** VUSs with conflicting data on ClinVar.

Gene	VUS	Reference Sequence	Uncertain Significance	Likely Benign
APC	c.5392A>G (p.Asn1798Asp)	rs200794097	yes	x
APC	c.8462A>G (p.Asp2821Gly)	rs780049836	x	x
APC	c.7399C>A (p.Pro2467Thr)	rs372305287	x	x
APC	c.5026A>G (p.Arg1676Gly)	rs370560998	x	x
ATM	c.8246A>T (p.Lys2749Ile)	rs779145081	x	x
ATM	c.6179G>A (p.Arg2060His)	rs376521407	x	x
ATM	c.6919C>T (p.Leu2307Phe)	rs56009889	x	x
ATM	c.670A>G (p.Lys224Glu)	rs145053092	x	x
ATM	c.7903A>G (p.Thr2635Ala)	rs886059799	x	x
AXIN2	c.1985T>C (p.Leu662Pro)	rs142476324	x	x
AXIN2	c.203G>A (p.Arg68Gln)	rs138056036	x	x
BARD1	c.556A>G (p.Ser186Gly)	rs16852741	x	x
BARD1	c.1360C>G (p.Pro454Ala)	rs730881408	x	x
BARD1	c.748T>C (p.Ser250Pro)	rs570022823	x	x
BARD1	c.1835A>T (p.Asp612Val)	rs201140528	x	x
BRCA1	c.3650C>G (p.Ser1217Cys)	rs398122676	x	x
BRCA1	c.2075A>C (p.His692Pro)	rs2053831947	x	x
BRCA1	c.4339C>A (p.Gln1447Lys)	rs1567868598	x	x
BRCA2	c.3318C>G (p.Ser1106Arg)	rs1298550035	x	x
BRCA2	c.343A>G (p.Lys115Glu)	rs56242644	x	x
BRIP1	c.226G>A (p.Val76Ile)	rs769573395	x	x
BRIP1	c.436A>G (p.Ile146Val)	rs1567868598	x	x
CHEK2	c.1111C>T (p.His371Tyr)	rs531398630	x	x
CHEK2	c.580A>T (p.Ser194Cys)	rs786203042	x	x
CHEK2	c.539G>T (p.Arg180His)	rs137853009	x	x
CHEK2	c.663C>G (p.Ile221Met)	rs200451612	x	x
DICER1	c.248A>G (p.Tyr83Cys)	rs373646414	x	x
GALNT12	c.907G>A (p.Asp303Asn)	rs145236923	x	x
GALNT12	c.1655G>A (p.Arg552Lys)	rs1285871027	x	x
MEN1	c.941G>A (p.Arg314Gln)	rs771645621	x	x
MSH2	c.1856A>G (p.Tyr619Lys)	rs63749982	x	x
MSH2	c.2606C>A (p.Ala869Glu)	rs730881772	x	x
MSH2	c.123C>G (p.Asp41Glu)	rs761960690	x	x
MSH2	c.118G>A (p.Gly40Ser)	rs63751260	x	x
MSH2	c.1489A>G (p.Ile497Val)	rs755501968	x	x
MSH2	c.157G>T (p.Ala53Ser)	rs755931648	x	x
MSH3	c.2732T>G (p.Leu911Trp)	rs41545019	x	x
MSH3	c.909G>C (p.Lys303Asn)	rs757164724	x	x
MSH3	c.582C>G (p.Asp194Glu)	rs749446559	x	x
MSH3	c.2732T>G (p.Leu911Trp)	rs41545019	x	x
MSH3	c.2173G>A (p.Glu725Lys)	rs200612739	x	x
MSH3	c.1361G>A (p.Arg454Gln)	rs144798521	x	x
MSH6	c.743G>A (p.Arg248Gln)	rs764870249	x	x
MSH6	c.4002-8dup	rs267608139	x	x
MSH6	c.3762A>T (p.Glu1254Asp)	rs375459388	x	x
MSH6	c.3762A>T (p.Glu1254Asp)	rs375459388	x	x
MUTYH	c.1255G>A (p.Ala419Thr)	rs587780744	x	x
NBN	c.1720T>A (p.Leu574Ile)	rs142334798	x	x
NBN	c.643C>T (p.Arg215Trp)	rs34767364	x	x
NBN	c.595C>T (p.Pro199Ser)	rs587780097	x	x
NBN	c.1343A>T (p.Gln448Leu)	rs146403088	x	x
NBN	c.511A>G (p.Ile171Val)	rs61754966	x	x
NF1	c.3883A>G (p.Thr1295Ala)	rs143836226	x	x
NF1	c.3315A>G (Silent)	rs1555614915	x	x
NF1	c.169G>A (p.Gly57Ser)	rs779727341	x	x
NHTL1	c.527T>C (p.Ile176Thr)	rs1805378	x	x
PALB2	210A>G (p.Ser70Ser)	rs786202650	x	x
PGDFRA	c.1425G>T (p.Glu475Asp)	rs200309940	x	x
PMS2	c.1510G>C (p.Glu504Gln)	rs368516768	x	x
PMS2	c.1004A>G (p.Asn335Ser)	rs200513014	x	x
POLD1	c.961G>A (p.Gly321Ser)	rs41554817	x	x
RAD50	c.1720A>C (p.Lys574Gln)	rs779597467	x	x
RAD50	c.1094G>A (p.Arg365Gln)	rs146370443	x	x
RAD51D	c.26G>C (p.Cys9Ser)	rs140825795	x	x
RNF43	c.1660C>T (p.Arg554Trp)	Variation ID: 1140674	x	x

For VUSs for which a reference sequence is not yet available, we provide the Variation ID from ClinVar (https://www.ncbi.nlm.nih.gov/clinvar/ (accessed on 26 November 2023)).

**Table 6 cancers-15-05762-t006:** VUS incidence rate in relation to age as well as personal and family history of cancer.

		All Patients N (%) *	Positive for VUS N (%) **	*p*-Value
Age at time of testing	≤40	204 (30.7%)	56 (27.4%)	0.730
	>40	459 (69.2%)	132 (28.7%)	
Ethnicity	Hispanic	35 (5.27%)	12 (34.2%)	0.374
	Non-Hispanic	378 (57.01%)	103 (27.2%)	
Race	White	333 (50.20%)	81 (24.3%)	**0.019**
	African-American	20 (3.01%)	8 (40%)	
	Asian	34 (5.12%)	15 (44.1%)	
Personal history of breast cancer	Yes	176 (26.5%)	62 (35.2%)	**0.018**
	No	487 (73.4%)	126 (25.8%)	
Family history of cancer in 1st-degree relative	Yes	351 (52.9%)	107 (30.4%)	0.253
	No	299 (45.1%)	79 (26.4%)	
Family history of breast cancer in 1st-degree relative	Yes	116 (17.4%)	44 (37.9%)	**0.018**
	No	394 (59.4%)	105 (26.6%)	
Family history of ovarian cancer in 1st-degree relative	Yes	23 (3.4%)	11 (47.8%)	**0.047**
	No	490 (73.9%)	140 (28.5%)	
Family history of prostate cancer in 1st-degree relative	Yes	31 (4.89%)	6 (19.3%)	0.203
	No	482 (72.6%)	145 (30.0%)	
Family history of colon cancer in 1st-degree relative	Yes	21 (3.1%)	4 (19.0%)	0.286
	No	492 (74.2%)	147 (29.8%)	

* Percentage refers to the proportion of patients out of the total number of patients included in this study. ** Percentage refers to the proportion of the patients positive for VUSs out of the total number of patients in the particular subgroup indicated in the variable column. Bold represents *p* < 0.05 and therefore significant result.

## Data Availability

Data are available upon request.

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
