# Peer review of "Prevalence of Variants of Uncertain Significance in Patients Undergoing Genetic Testing for Hereditary Breast and Ovarian Cancer and Lynch Syndrome"

_cancers, 2023, doi:10.3390/cancers15245762_

Round 1

Reviewer 1 Report

Comments and Suggestions for Authors

Introduction.

In the introduction HBOC and Lynch sy connection to cancer are differently described.

Materials and Methods.

Genetic testing

Line 84, 87, 92 - Explanation of abbreviation lake EPIC, PACS, MRNs, IRB…  is missing and should be added. What are EPIC, Meditech, PACS ? – hospital electronic records about patients data?

Line 114-115. The decision - “We considered variants as positive if at least one pathogenic report in ClinVar. “ might lead to overestimation of pathogenicity of assessed variants and would be suggested to discuss it in discussion. What is the percentage of pathogenic variants with conflicting evidence? Additional remark; did you mean “We consider result as positive…”

Were all pathogenic variants and VUS variants described in CliVar? Were the patients with variants not found in ClinVar excluded from analysis? – this should be described here.

The gene panels for which the patients were tested, are missing. Were all the patients tested for the same gene panel? Did the gene panel contain only the genes responsible for HBOC and Lynch (described in NCCN) or also other? – this should be added in Materials and Methods section

There is no explanation, if the VUSs counted in this study, were only from NCCN suggested HBOC genes and Lynch genes? Are they are taken for broader cancer associated genes? Authors should state in what kind of gene panel they counted VUS, or patients with VUS, and what is rational behind it.- this should be added in Materials and Methods section

In the supplementary table 1 – Some patients listed there, as having VUS, have VUS in non-HBOC and non-Lynch genes (at least not-according to NCCN guidelines), for example: PDGFA, RECQL4, SMARCE1, DICER1, CDKN2A, CDK4 … are those also counted as VUS positive patients?

Results.

Lines 172-173.- “ ..the most common recommendations for VUS were annual MRI…” – where was this observed from?  From electronic records? - is there any statistics on this? is this true for any VUS (or only those in NCCN HBOC and Lynch genes)?

Explanation of abbreviations is missing.

In the Table 3 there is no explanation, what this percentage refers to.

Discussion.

Lines 192-193. - Authors should reconsider the statement “… MUTYH is one of least frequent genes associated with Lynch syndrome..” – since this is not correct.  Bialelic pathogenic variants in MUTYH are associated with “MUTYH associated polyposis” (adenomatous) and not with Lynch syndrome (NCCN gudelines - Genetic/Familial High-Risk Assessment: Colorectal).-

The article that the authors are referring to (Castillejo et al 2014) states only “MUTYH-associated polyposis patients show some phenotypic similarities to Lynch syndrome patients.”  And state that bialelic mutations in MUTYH might explain some cases considered to be Lynch sy like, but does not address MUTYH as being gene associated with Lynch syndrome.

Line 210-212 – Sentence “… VUS prevalence was unrelated to Hispanic ethnicity but was related to race…” – it is not very clear. What is post-hoc analysis? -statistical?

Is it possible that in non-white population – screening and germline testing is less available, subsequently we have less information concerning rare variants (les population-based studies) and consequently there are more variants that stay in VUS and cannot be classifies as likely benign or benign variants?

In the discussion the consideration on reclassification on the basis of ACMG/AMP classification guidelines or even more precise CliGene or CanVIG  gene specific guidelines, which might lower the proportion of VUS, should also be considered.

ACMG /AMP gudelines: Richards, S. et al. (2015) 'Standards and guidelines for the interpretation of sequence variants: a joint consensus recommendation of the American College of Medical Genetics and Genomics and the Association for Molecular Pathology.', Genetics in medicine: official journal of the American College of MedicalGenetics, 17(5), pp. 405-424. doi: 10.1038/gim.2015.30.

https://clinicalgenome.org/working-groups/sequence-variant-interpretation/

https://www.cangene-canvaruk.org/canvig-uk-guidance

Supplementary table.

Reference sequences should be listed for all genes, where VUS were found.

The patient subsequent number of is suggested to be added in the supplementary table. Abbreviation used in the table should be explained in the table.

References.

The reference no 25. Arakane, M (Synthesis and evaluation of polyfunctional monomers with acrilat…” – doesn’t seem to be appropriate for this article.- the authors should delete this reference.

Author Response

Reviewer - 1

Introduction.

In the introduction HBOC and Lynch sy connection to cancer are differently described.

Response: We have now added section on overlap between these two syndromes.

“As our understanding of hereditary cancer syndromes advances, a noteworthy convergence has emerged among these syndromes. This intersection is particularly evident in the increased risks associated with certain cancers. Both Pancreatic and ovarian cancers, for instance, exhibit heightened risks in hereditary cancer syn-dromes(10-13). Lynch Syndrome (LS) alone accounts for 13-15% of ovarian cancer cases(14). Literature also underscores the association between specific mutations and an elevated risk of certain cancers. BRCA1 and BRCA2 mutations, for instance, are linked to increased risks of Endometrial and colorectal cancers(15), while Lynch syn-drome is associated with a higher breast cancer risk(16). Notably, Lynch syndrome pa-tients with specific mutations, such as PMS2 and those exhibiting MMR deficiency, face an elevated risk of developing breast cancer (17, 18). The accumulating evidence designates breast cancer as an extra colonic manifestation of Lynch syndrome(17, 18). Furthermore, the co-occurrence of pathogenic variants in both Hereditary Breast and Ovarian Cancer Syndrome (HBOC) and Lynch Syndrome is on the rise, underscoring the complex interplay between these genetic predispositions(19)..”

Materials and Methods.

Genetic testing

Line 84, 87, 92 - Explanation of abbreviation lake EPIC, PACS, MRNs, IRB…  is missing and should be added. What are EPIC, Meditech, PACS ? – hospital electronic records about patients data?

Response: Thank you for bringing this to our attention. We have provided the requested abbreviations. For “EPIC”, which refers to the healthcare software system, there was no acronym identified.

Line 114-115. The decision - “We considered variants as positive if at least one pathogenic report in ClinVar. “ might lead to overestimation of pathogenicity of assessed variants and would be suggested to discuss it in discussion. Additional remark; did you mean “We consider result as positive…”

Response: We have restructured our section as follows:

Genetic mutation information was extracted from patients' genetic testing laboratory reports. All our patients underwent testing via the commercial Invitae or Myriad genetic panels. These panels encompassed a comprehensive 20+ gene hereditary cancer panel, inclusive of both HBOC and Lynch genes, along with several others (Supplementary Material). Patient records were collected for the period spanning 2006 to 2020. Notably, genetic panels underwent modifications over time, resulting in variations among tests. These changes were influenced by the addition of new genes or, in some instances, patients opting for more limited panels based on their preferences or insurance coverage. Our analysis considered all Variants of Uncertain Significance (VUS) detected through genetic testing, extending beyond the confines of NCCN-designated HBOC and Lynch syndrome genes. For result interpretations, re-ports without identified mutations were categorized as "negative." Conversely, if a mutation was identified, it was classified as "positive" (indicating a pathogenic mutation), "VUS," or grouped with the negative results if deemed benign.

To classify mutations, we utilized ClinVar 19, designating results as "positive" if they had at least one pathogenic report in ClinVar. Variants were considered VUS if they lacked pathogenic reports in ClinVar but had at least one report of unclear significance. Variants were designated as negative if they lacked pathogenic or unclear significance reports in ClinVar. The determination of results as positive, negative, or VUS was based on ClinVar reports published as of July 9, 2023. Mutations not previously reported in ClinVar were considered VUS and were included in our analysis.”

We have also added following sentences in our discussion:

“ Of note, some of the assessed variants had only one pathogenic report increasing to uncertainty. In our study, we considered variants as positive if at least one pathogenic report was identified in ClinVar which can lead to over-estimation of pathogenicity of assessed variants.”

What is the percentage of pathogenic variants with conflicting evidence?

Response: All pathogenic mutations were reported classified as pathogenic in ClinVar, and among them, 12 out of 90 (13.3%)  had conflicting reports indicating uncertain significance.

Were all pathogenic variants and VUS variants described in CliVar? Were the patients with variants not found in ClinVar excluded from analysis? – this should be described here.

Response: All pathogenic mutations were described in ClinVar. Eleven mutations were not previously reported in ClinVar or to any other source of our knowledge. We considered those 11 mutations to be VUS and included them in our analysis. We have updated the Methods and Results sections accordingly. A list of the these eleven mutations is provided in Table 4.

The gene panels for which the patients were tested, are missing. Were all the patients tested for the same gene panel? Did the gene panel contain only the genes responsible for HBOC and Lynch (described in NCCN) or also other? – this should be added in Materials and Methods section

Response: We have added a detailed explanation in the Material & Methods section as well as in supplementary material. Patients tested through the commercial Invitae or Myriad genetic panels. Patients had a 20+ gene hereditary cancer panel that included both HBOC and Lynch genes as well as several others. Genetic panels have changed over the follow-up period of this study (2006-2020) and therefore patients did not have the exact same test. New genes were added, or some patients had more limited panels based on their preference or insurance coverage. We added in the methodology a list of the 20 genes that the vast majority of patients were typically tested for at minimum. If deemed needed by the reviewers, we can go back to the medical records and extract the exact genes for which each patient was tested. In that case, we would need a more extended period to revise the manuscript.

There is no explanation, if the VUSs counted in this study, were only from NCCN suggested HBOC genes and Lynch genes? Are they are taken for broader cancer associated genes? Authors should state in what kind of gene panel they counted VUS, or patients with VUS, and what is rational behind it.- this should be added in Materials and Methods section

Response: We have enhanced the clarity of our Methods section by explicitly stating that all Variants of Uncertain Significance (VUS) detected through genetic testing were incorporated into our analysis. Unlike limiting our analysis to the National Comprehensive Cancer Network (NCCN)-recommended genes for Hereditary Breast and Ovarian Cancer (HBOC) and Lynch syndrome, we opted for a broader approach. Our rationale was to comprehensively assess the overall burden of VUS results. Considering that VUS signifies uncertain significance, it may not yield a significant clinical impact by restricting our results solely to HBOC and Lynch genes. We believe that a broader approach provides a more comprehensive understanding of the implications of VUS across various genes.

In the supplementary table 1 – Some patients listed there, as having VUS, have VUS in non-HBOC and non-Lynch genes (at least not-according to NCCN guidelines), for example: PDGFA, RECQL4, SMARCE1, DICER1, CDKN2A, CDK4 … are those also counted as VUS positive patients?

Response: Yes, we included all VUS that were detected with the genetic testing, not restricted to NCCN HBOC and Lynch syndrome genes. As explained above, the genetic panel varied among each patient.

Results.

Lines 172-173.- “ ..the most common recommendations for VUS were annual MRI…” – where was this observed from?  From electronic records? - is there any statistics on this? is this true for any VUS (or only those in NCCN HBOC and Lynch genes)?

Response: Information about follow-up recommendations were collected as all our other data through electronic medical records. The results refer to any VUS (not limited to the HBOC and Lynch syndrome genes only). Percentages reflecting proportions of each recommendation are now provided in the results section.

Explanation of abbreviations is missing.

Response: We have provided the missing explanation for the abbreviations used in the Results section (MAH, MRI, PCP).

In the Table 3 there is no explanation, what this percentage refers to.

Response: Thank you, we provided explanation of the proportions to which the percentages refer to.

Discussion.

Lines 192-193. - Authors should reconsider the statement “… MUTYH is one of least frequent genes associated with Lynch syndrome..” – since this is not correct.  Bialelic pathogenic variants in MUTYH are associated with “MUTYH associated polyposis” (adenomatous) and not with Lynch syndrome (NCCN gudelines - Genetic/Familial High-Risk Assessment: Colorectal). The article that the authors are referring to (Castillejo et al 2014) states only “MUTYH-associated polyposis patients show some phenotypic similarities to Lynch syndrome patients.”  And state that bialelic mutations in MUTYH might explain some cases considered to be Lynch sy like, but does not address MUTYH as being gene associated with Lynch syndrome.

Response: We appreciate your valuable observation, and based on your suggestion, we have revised our manuscript accordingly:

“In contrast, MUTYH is primarily associated with MUTYH-associated polyposis exhib-iting some phenotypic similarities to those with Lynch syndrome. In a European-based study, MUTYH accounted for only 3.6% of Lynch syndrome-like cases. In our study, however, we identified MUTYH pathogenic mutations in 13.2% of cases. This disparity may be attributed to incidental factors, population-specific genetic variability, or other contributing factors.”

Line 210-212 – Sentence “… VUS prevalence was unrelated to Hispanic ethnicity but was related to race…” – it is not very clear. What is post-hoc analysis? -statistical?

Response: We appreciate your feedback, and we have made the following clarifications in our manuscript:

"In addition, in our study, VUS prevalence showed no significant difference between the Hispanic and non-Hispanic populations. However, when examining different racial groups, we observed significantly higher VUS rates in all non-white groups. A sub-group analysis further revealed that the significant difference in VUS rates was specifically observed between the Asian and White populations."

Is it possible that in non-white population – screening and germline testing is less available, subsequently we have less information concerning rare variants (les population-based studies) and consequently there are more variants that stay in VUS and cannot be classifies as likely benign or benign variants?

In the discussion the consideration on reclassification on the basis of ACMG/AMP classification guidelines or even more precise CliGene or CanVIG  gene specific guidelines, which might lower the proportion of VUS, should also be considered.

ACMG /AMP gudelines: Richards, S. et al. (2015) 'Standards and guidelines for the interpretation of sequence variants: a joint consensus recommendation of the American College of Medical Genetics and Genomics and the Association for Molecular Pathology.', Genetics in medicine: official journal of the American College of MedicalGenetics, 17(5), pp. 405-424. doi: 10.1038/gim.2015.30.

https://clinicalgenome.org/working-groups/sequence-variant-interpretation/

https://www.cangene-canvaruk.org/canvig-uk-guidance

Response: Thank you for your recommendation. We have incorporated a dedicated section describing the guidelines in our discussion.

Supplementary table.

Reference sequences should be listed for all genes, where VUS were found.

Response: Due to the importance of Supplementary tables for our manuscript, we have moved those tables as the main tables in the manuscript.

We have added the reference sequences for all VUS in Table 3. For some VUS, a reference sequence is not yet available and thus the Variation ID from ClinVar was provided instead. We have included the VUS that have not been previously reported in ClinVar in Table 4.

The patient subsequent number of is suggested to be added in the supplementary table. Abbreviation used in the table should be explained in the table.

Response: We apologize but the first part of this request is not clear to the authors. Should we add the patient study ID? For the second part, we updated the table to include explanations of the abbreviations.

References.

The reference no 25. Arakane, M (Synthesis and evaluation of polyfunctional monomers with acrilat…” – doesn’t seem to be appropriate for this article.- the authors should delete this reference.

Response: Thank you for bringing this to our attention. We have deleted the reference.

Reviewer 2 Report

Comments and Suggestions for Authors

In their study, the authors investigated the VUS rate in a cohort of 663 hospital-based cases from families with hereditary tumor diseases that met the stated criteria for genetic testing. The occurrence of VUS was examined retrospectively and stratified by subgroup. The results are described clearly and comprehensibly. The study thus provides a good overview of the expected VUS rate. However, the descreptive approach remains, a truly scientific evaluation is missing. Some points should therefore be addressed in any case. 

1. Are there any differences in terms of screening with prediction programs between VUS found in cases with personal and/or family history compared to those found in cases without personal or family history? Is there evidence of an accumulation of predicted potentially harmful variants in the affected cases?

2. What kind of variants are involved in the mutations found predominantly in Asians? Are these variants also found in population-based databases predominantly in Asians and therefore possibly poorly characterized so far?

3. What are the recommendations for dealing with VUS and patients who have a VUS? The assessment of VUS variants can change, so there may be the possibility of a recall. Should only centers with experience in dealing with VUS perform genetic testing. What about sharing data with expert groups for variant classification like the ENIGMA or INSIGHT consortium? Should this be a prerequisit for panel testing instead of just using given ClinVar classifications or results from Myriad/Invitae.

Author Response

Reviewer 3:

In their study, the authors investigated the VUS rate in a cohort of 663 hospital-based cases from families with hereditary tumor diseases that met the stated criteria for genetic testing. The occurrence of VUS was examined retrospectively and stratified by subgroup. The results are described clearly and comprehensibly. The study thus provides a good overview of the expected VUS rate. However, the descreptive approach remains, a truly scientific evaluation is missing. Some points should therefore be addressed in any case. 

1.Are there any differences in terms of screening with prediction programs between VUS found in cases with personal and/or family history compared to those found in cases without personal or family history? Is there evidence of an accumulation of predicted potentially harmful variants in the affected cases?

Response: Screening prediction models for VUS and recommendations was not withing our objectives. However, our data showed that patients with personal or family history of breast or ovarian cancer were more likely to get a VUS report compared to their counterparts. Interestingly, family history of cancer overall or family history of prostate or colon cancer did not appear to be associated with VUS detection.

  1. What kind of variants are involved in the mutations found predominantly in Asians? Are these variants also found in population-based databases predominantly in Asians and therefore possibly poorly characterized so far?

Response: ATM and MSH6 were the most common genes with VUS in the Asian population. In the African American population, the most common genes were NBN and APC. Notably, these genes were encompassed within the 10 most common genes with VUS in the overall cohort, suggesting no inherent gene predominance based on ethnicity. However, as we explain in our discussion, lack of ancestral variety in genetic databases contributes to a certain degree of inadequacy in characterizing mutations in individuals of Asian or African American descent, consequently impacting classification accuracy. We have also included the ACMG guidelines in our discussion that highlight the pivotal role of demographic distribution in determining the pathogenic or benign nature of a variant.

  1. What are the recommendations for dealing with VUS and patients who have a VUS? The assessment of VUS variants can change, so there may be the possibility of a recall. Should only centers with experience in dealing with VUS perform genetic testing. What about sharing data with expert groups for variant classification like the ENIGMA or INSIGHT consortium? Should this be a prerequisit for panel testing instead of just using given ClinVar classifications or results from Myriad/Invitae.

Response: Thank you for your comment. It served as valuable feedback that encouraged us to enhance our discussion and elaborate on recommendations for clinicians dealing with VUS results and current initiatives aimed at resolving the challenges associated with VUS.

Reviewer 3 Report

Comments and Suggestions for Authors

Please find my comments and suggestions below.

First of all, VUS in which genes were studied? The authors provide info regarding most common genes in which VUS were detected (Figure 1). Did they screen a panel of genes, or whole genome sequencing data?

Line 54. The authors state “The two most common hereditary cancer syndromes are Hereditary Breast and Ovarian Cancer Syndrome (HBOC) and Lynch Syndrome”. There is an ongoing debate about possible interconnection of these two syndromes (in other words, whether Lynch syndrome includes HBOC-associated tumours as well) supported by body literature and experimental data. Please discuss in detail and refer to the literature. Only 30 references provided in this article is definitely not enough to cover the subject. The literature review should be expanded significantly and at least 20 more references should be added, also with focus on ethnic and race component of the genetic of HBOC and Lynch syndrome.

Please specify what exact VUS were determined for each gene (as a supplementary table perhaps) – the type of sequence alteration, its location/coordinates, etc. It will be instrumental for other researchers working in the field.

As for the VUS detected in your study, can you compare their frequencies to the previously reported for healthy population of the similar ethnic composition? What about possibility of re-interpretation of some of the VUS detected in your study, based on their distribution in healthy vs Lynch syndrome and HBOC cohorts (based on further investigations in future)? (as the authors state in line 226, perhaps some of these VUS might be later reclassified as pathogenic“) Please discuss in detail

Line 224. The authors state “Furthermore, we show that individuals with a personal history of breast cancer or family history of breast/ovarian cancer are more likely to have a VUS”. More likely compared to whom? Please specify.

Line 109. Is there any chance to deposit de-identified raw sequencing data for the cohort to the NCBI database?

Line 145. in the table with patients characteristics, in the “Race” and “Ethnicity” columns please provide info about gender for each race and ethnicity.

Author Response

Please find my comments and suggestions below.

First of all, VUS in which genes were studied? The authors provide info regarding most common genes in which VUS were detected (Figure 1). Did they screen a panel of genes, or whole genome sequencing data?

We have provided a comprehensive explanation in both the Material & Methods section and supplementary material. Patients underwent testing through the commercial Invitae or Myriad genetic panels, which consisted of a 20+ gene hereditary cancer panel encompassing HBOC and Lynch genes, among others. Notably, the composition of genetic panels evolved during the study's follow-up period (2006-2020), resulting in variations among tests conducted.Patients did not undergo identical tests due to these changes, with some individuals having additional genes included, while others had more limited panels based on personal preference or insurance coverage. To offer insight into the commonly tested genes, we included a supplementary list of the 20 genes that the majority of patients were typically tested for at a minimum.

Line 54. The authors state “The two most common hereditary cancer syndromes are Hereditary Breast and Ovarian Cancer Syndrome (HBOC) and Lynch Syndrome”. There is an ongoing debate about possible interconnection of these two syndromes (in other words, whether Lynch syndrome includes HBOC-associated tumours as well) supported by body literature and experimental data. Please discuss in detail and refer to the literature. Only 30 references provided in this article is definitely not enough to cover the subject. The literature review should be expanded significantly and at least 20 more references should be added, also with focus on ethnic and race component of the genetic of HBOC and Lynch syndrome.

Response: We appreciate your suggestion for adding evidence to the overplay between HBOC and Lynch syndrome. We have added the following paragraph to Introduction:

“As our understanding of hereditary cancer syndromes advances, a noteworthy convergence has emerged among these syndromes. This intersection is particularly evident in the increased risks associated with certain cancers. Both Pancreatic and ovarian cancers, for instance, exhibit heightened risks in hereditary cancer syndromes (references: 26315041, 19383374, 23099806, 16360201). Lynch Syndrome (LS) alone accounts for 13-15% of ovarian cancer cases (reference: 25279173). Literature also underscores the association between specific mutations and an elevated risk of certain cancers. BRCA1 and BRCA2 mutations, for instance, are linked to increased risks of Endometrial and colorectal cancers (reference: 18398828), while Lynch syndrome is associated with a higher breast cancer risk (reference: 23562522). Notably, Lynch syndrome patients with specific mutations, such as PMS2 and those exhibiting MMR deficiency, face an elevated risk of developing breast cancer (references: .32285031, 22034109). The accumulating evidence designates breast cancer as an extra colonic manifestation of Lynch syndrome. (references: .32285031, 22034109) Furthermore, the co-occurrence of pathogenic variants in both Hereditary Breast and Ovarian Cancer Syndrome (HBOC) and Lynch Syndrome is on the rise, underscoring the complex interplay between these genetic predispositions (reference: 36232793).”

Please specify what exact VUS were determined for each gene (as a supplementary table perhaps) – the type of sequence alteration, its location/coordinates, etc. It will be instrumental for other researchers working in the field.

We provide this information in Supplementary Table 1.

As for the VUS detected in your study, can you compare their frequencies to the previously reported for healthy population of the similar ethnic composition? What about possibility of re-interpretation of some of the VUS detected in your study, based on their distribution in healthy vs Lynch syndrome and HBOC cohorts (based on further investigations in future)? (as the authors state in line 226, perhaps some of these VUS might be later reclassified as pathogenic“) Please discuss in detail

Response: As per our literature search, most studies in VUS rates have been performed retrospectively in patients that underwent genetic testing due to family or personal history. Thus, comparing with healthy adults based on the current literature might be challenging. To the second part of your point, we have included a paragraph in our discussion dedicated on the ACMG guidelines on sequence classification, where we are also discussing in detail the possibility of sequence reclassification based on the frequency in patients with cancer syndrome vs healthy counterparts.

Line 224. The authors state “Furthermore, we show that individuals with a personal history of breast cancer or family history of breast/ovarian cancer are more likely to have a VUS”. More likely compared to whom? Please specify.

We have provided an explanation to the comparison.

Line 109. Is there any chance to deposit de-identified raw sequencing data for the cohort to the NCBI database?

If needed, we are happy to deposit the de-identified raw sequencing data to the NCBI database. 

Line 145. in the table with patients characteristics, in the “Race” and “Ethnicity” columns please provide info about gender for each race and ethnicity.

We added gender information for race and ethnicity in Table 1 and also provided some additional explanations at the table footnote.

Round 2

Reviewer 2 Report

Comments and Suggestions for Authors

My comments were taken into consideration.

Reviewer 3 Report

Comments and Suggestions for Authors

All my comments have been addressed by the authors, thank you